The benefits of social insurance system prediction using a hybrid fuzzy time series method

Khalil Ahmed Abdelreheem ahmedrohim91@aun.edu.eg 1 2
Mandour Mohamed Abdelaziz 3
Ali Ahmed 4 5
1 School of Mathematics and Statistics, Central South University , Changsha , Hunan , China
2 Faculty of Commerce, Assiut University , Assiut , Egypt
3 Department of Business Administration, Faculty of Commerce, Mansoura University , Mansoura , Egypt
4 Department of Computer Science, College of Computer Engineering and Sciences, Prince Sattam Bin Abdulaziz University , Al-Kharj , Saudi Arabia
5 Higher Future Institute for Specialized Technological Studies , Cairo , Egypt
Alatas Bilal
Electronic publication date: 2024 Nov 26
Publication date: 2024
Volume: 10
Electronic Location ID: e2500
Received 2024 Jul 23; Accepted 2024 Oct 21
Copyright: ©2024 Khalil et al.
Copyright year: 2024
Copyright holder: Khalil et al.
License: This is an open access article distributed under the terms of the Creative Commons Attribution License, which permits unrestricted use, distribution, reproduction and adaptation in any medium and for any purpose provided that it is properly attributed. For attribution, the original author(s), title, publication source (PeerJ Computer Science) and either DOI or URL of the article must be cited.
License URL: https://creativecommons.org/licenses/by/4.0/

Keywords: Fuzzy time series, Markov chain, Predictive method, Insurance, Forecasting

Funding: the Prince Sattam bin Abdulaziz University PSAU/2024/R/1446 This study is funding by the Prince Sattam bin Abdulaziz University project number (PSAU/2024/R/1446). The funders had no role in study design, data collection and analysis, decision to publish, or preparation of the manuscript.

==============================
Decision-making in many industries relies heavily on accurate forecasts, including the insurance sector. The Social Insurance System (SIS) in Egypt, operating under a fully funded paradigm, depends on reliable predictions to ensure effective financial planning. This research introduces a hybrid predictive model that combines fuzzy time series (FTS) Markov chains with the tree partition method (TPM) and difference transformation to forecast total pension benefits within Egypt’s SIS. A key feature of the proposed model is its ability to optimize the partitioning process, resulting in the creation of nine intervals that reduce computational complexity while maintaining forecasting accuracy. These intervals were consistently applied across all fuzzy time series models for comparison. The model’s performance is evaluated using established metrics such as MAPE, Thiels’ U statistic, and RMSE. Additionally, prediction interval coverage probability (PICP) and mean prediction interval length (MPIL) are used to assess the quality of prediction intervals, with a 95% prediction interval serving as the baseline. The proposed model achieved a PICP of approximately 95%, indicating well-calibrated prediction intervals, although the MPIL of 424.5 reflects a wider uncertainty range. Despite this, the model balances coverage accuracy and interval precision effectively. The results demonstrate that the proposed model significantly outperforms traditional models like linear regression, ARIMA, and exponential smoothing and conventional FTS models like Song, Chen, Yu, and Cheng by achieving the lowest MAPE with the value of 11.8% for training and 10.65% for testing. This superior performance highlights the model’s reliability and potential applicability to further forecasting tasks in the field of insurance and beyond.

Introduction

Forecasting is a critical aspect for solving problems, particularly in various businesses and the insurance industry, where predictions of future events must inform decision making. Therefore, forecasting techniques enable the insurance authorities to anticipate future results and formulate strategies accordingly. Thus, accurate and efficient forecasting systems are highly sought after in this industry (Khalil, Liu & Ali, 2022a).

The social insurance system (SIS) plays a pivotal role in countries around the world as a cornerstone of social welfare and economic stability. Its significance lies in its ability to provide a safety net for individuals and families during times of need, ensuring that unforeseen events such as illness, unemployment, or disability do not lead to financial ruin (Liu et al., 2023; Raouf & Elsaieed, 2022). Consequently, the SISs serve as a means for individuals to actively participate in safeguarding their future. Thus, these systems rely on contributions from both workers and employers, effectively distributing risks among a wide population. By doing so, social insurance programs promote social cohesion and mitigate income disparities, thereby cultivating a sense of security and well-being among the populace (Chen et al., 2022).

SIS holds profound significance in countries as a critical component of their economic and social fabric. Additionally, these programs often stimulate economic growth by maintaining consumer spending even in times of economic downturns, as individuals have a safety net to fall back on. Moreover, they contribute to the overall health and productivity of the workforce, as access to healthcare and unemployment benefits can prevent individuals from falling into poverty traps. In essence, social insurance embodies a nation’s commitment to the welfare and resilience of its citizens, promoting economic stability and social justice. Simultaneously, it is widely acknowledged as a fundamental issue that numerous nations grapple with due to the intricate interplay between this matter and its implications on public finances (Raouf & Elsaieed, 2022).

In Egypt, SIS operates on a fully funded model where employee contributions are invested for pension payouts, evolving over time to include elements of a defined benefit plan. Governed by laws, the latest being Law 148/2019, this legal framework aims to modernize the system. However, Egypt’s SIS faces challenges due to fragmentation and diverse benefit packages for different workforce segments, covering areas like old age, disability, and illness. Unsustainability and inefficiency arise from low real interest rates for pension fund investments, generous pensions, and flexible pension adjustments based on recent contributions. Many workers under-report their income throughout most of their working careers to reduce their payments, threatening the system’s financial viability (Kassem, 2021; Loewe & Westemeier, 2018). In light of Egypt’s significant and increasing total benefits (pension), which have exhibited a notable upward trajectory in recent years, there is a need to forecast these benefits for the coming years.

Actuarial models are employed to estimate annual benefits, forming the basis for demographic and financial predictions in pension systems. These models rely on factors such as demographics and economics and are vital for supporting policy decisions. Consequently, the integration of actuarial methodologies in tandem with predictive analytics assumes paramount importance in comprehending and forecasting future behavioral trends or occurrences, thereby furnishing the requisite support for informed policy formulation and decision-making. In essence, the prediction and evaluation of benefits have utmost significance for actuaries in their capacity as advisors for strategic decision-making within the realm of social insurance (Hassani, Unger & Beneki, 2020; Raouf & Elsaieed, 2022).

In our study, we utilized a summative modeling approach to predict retirement benefits, primarily based on historical data trends within the Egyptian social insurance system. Unlike traditional actuarial methodologies, which often employ mortality projections, such as cohort-based mortality tables to estimate future pension payments, our method focused on the available data. Actuarial literature, particularly from the Government Actuary’s Department and other studies, highlights the importance of mortality assumptions for accurately forecasting long-term pension liabilities. These actuarial models predict benefit changes by incorporating life expectancy data, where pension growth is typically modeled multiplicatively, reflecting the natural increase in benefit payouts as retirees age and longevity improves (Booth & Tickle, 2008).

However, due to the unavailability of detailed mortality data for Egypt’s population, we were unable to apply these traditional methods. Instead, we relied on a summative model, in which retirement benefits are predicted based on observed incremental changes in benefits, rather than factoring in mortality or life expectancy. Although our approach may seem basic by comparison, it offers a practical solution given the constraints of the available data. Incorporating more complex actuarial models in future work, including mortality rates, could provide a more refined projection of retirement benefits (Belloni & Maccheroni, 2013; Ingale & Paluri, 2023).

The study’s motivation and contribution

Fuzzy time series (FTS) models have exhibited more effectiveness in comparison to traditional forecasting methods in the field of forecasting, particularly when dealing with datasets that are incomplete, ambiguous, or have a limited number of observations (Khalil, Liu & Ali, 2024a). Furthermore, these models do not require the imposition of statistical assumptions. They use mathematical reasoning to build models and forecast future events based on historical linguistic data. In contrast to the traditional approaches taken for predicting time series, the inherent unpredictability of temporal observations is acknowledged and accounted for by fuzzy time series forecasting techniques. This eliminates the requirement for strict assumptions and a substantial amount of prior knowledge about the observed data. Moreover, it is important to highlight that fuzzy time series forecasting techniques demonstrate competence in handling a restricted number of data, emphasizing their flexibility and usefulness (Mao & Xiao, 2019; Wu et al., 2021).

In recent times, there has been a notable proliferation in the adoption of hybrid models as a means to enhance predictive accuracy. It has been demonstrated that the utilization of hybrid approaches frequently results in enhanced performance outcomes. The fundamental concept underlying hybridization models is their ability to synergistically enhance the identification of patterns in data. Numerous studies, both theoretical and empirical, have provided evidence supporting the notion that combining multiple models through hybridization leads to enhanced performance as compared to the utilization of a single model in isolation (Chen & Chen, 2015; Hadwan et al., 2022; Khairalla & Al-Jallad, 2017; Khozani et al., 2022).

Numerous fuzzy time series models have been advanced for predictive purposes across diverse domains. Building upon the merits inherent in these models and their demonstrated efficacy in predicting, this study presents a hybrid FTS that builds on the strengths of the fuzzy time series Markov chain (FTSMC), the difference transformation data technique, and the Tree Partition technique (TPM) for making predictions. The primary aim of this model is to forecast the total benefits (pensions) of the social insurance system in Egypt, reflecting a commitment to harness the capabilities of advanced methodologies for enhancing predictive accuracy in this specific application domain. The research presents several significant contributions to literature as outlined below:

1- The efficacy of FTS in modeling non-linear, with inherent uncertainties and accommodating linguistic variables, and providing adequate performance with limited data availability. Thus, this study investigates a hybrid prediction model that combines fuzzy time series with Markov chain, by leveraging the tree partition method (TPM) and data transformation method to enhance the accuracy of forecasting the total benefits (pensions) of the social insurance system in Egypt. Three metrics are proposed to evaluate the prediction performance of the model. This method can enhance the accuracy and effectively manage challenges in insurance data, such as uncertainty and limited dataset size, and assist actuaries in fulfilling their professional obligation to engage in strategic decision-making inside insurance businesses.

2- To the best of our knowledge, this work is the first empirical investigation to employ a hybrid FTS that builds on the strengths of the fuzzy time series Markov chain (FTSMC), the difference transformation data technique, and the tree partition technique (TPM) for predicting the total benefits (pensions) of social insurance system in Egypt. This research addresses a large gap in existing insurance literature, making it a valuable contribution to the field.

3- Finally, the author’s primary objective is to provide insurance authorities with a thorough and practical technique that can assist actuaries in fulfilling their professional duty of making strategic decisions within the realm of the social insurance system. Overall, the study offers several important contributions to the field of insurance and gives crucial insights into the Egyptian social insurance system.

The remainder of this work is outlined as follows: ‘Background and Literature Review’ provides a brief introduction to prior studies. ‘Data & Evaluation Methods’ provides a detailed account of the research methodology, including the research design, describe the specific techniques and approaches that were employed in the study and the evaluation metrics. The findings are presented and analyzed in ‘Results and Discussion’. Finally, we present the conclusion in ‘Conclusions’.

Background and Literature Review

Forecasting within the domain of insurance holds significant importance, necessitating the application of various statistical and soft computing methodologies. These methods play a pivotal role in the prediction of events and trends within the insurance industry. Enhanced forecasting capabilities are instrumental in aiding actuaries in their professional responsibilities, enabling the informed and strategic decision-making processes inherent to the management of social insurance systems. For instance, Ying et al. (2017) present a deep learning based RNN framework for forecasting payment behaviour using real data from Taiwan’s Ministry of Health and Welfare. This model outperforms SVM and HMM in various settings, although it addresses individual payment predictions, which differs from our research focus. Khalil, Liu & Ali (2022a) tried to forecast the loss ratios in petroleum insurance using both ANFIS and ARIMA models. They found that the ANFIS models provided more accurate results than the ARIMA models.

Fuzzy time series methods

In recent years, an increasing number of researchers have employed fuzzy time series methodologies to address prediction challenges as in Alyousifi et al. (2020), Alyousifi et al. (2021), Arnita, Afnisah & Marpaung (2020), Jilani, Burney & Ardil (2008), Lee et al. (2017), Mao & Xiao (2019), Rahim et al. (2018), Raouf & Elsaieed (2022), Tsaur (2012) and Wu et al. (2021). These fuzzy time series models have demonstrated notable efficacy when juxtaposed with conventional forecasting approaches, particularly within the realm of forecasting methodology. Tsaur (2012) designed a new model that integrates Fuzzy Time Series analysis with Markov chain simulation, referred to as FTSMC. This model was devised to ascertain suitable weights for fuzzy relationships within time series patterns, thereby enhancing model accuracy. However, FTSMC exhibits a limitation as it relies on arbitrary partitioning of intervals and does not possess the capability to determine the optimal interval length. This limitation constitutes a drawback in the context of effectively determining interval length within this hybrid model.

Fuzzy time series in insurance and actuarial science

Fuzzy time series (FTS) has become an invaluable tool in insurance and actuarial science, particularly for dealing with the uncertainty, vagueness, and imprecision inherent in many actuarial datasets. Actuaries, who traditionally rely on precise statistical methods, often face challenges when working with datasets where variability and ambiguity prevail common in areas like mortality rates, claims forecasting, and pension liabilities. The integration of fuzzy logic within time series models offers a flexible and powerful approach to overcoming these challenges.

Several studies have employed FTS methods to address various predictive challenges in insurance and actuarial science, including car accidents, profitability, insurance claims forecasting, and mortality rates. For example, Jilani, Burney & Ardil (2008) developed a multivariate fuzzy time series model for predicting car accidents in Belgium. Their approach demonstrated superior performance compared to existing techniques, offering actuaries a robust tool for forecasting in both car and life insurance contexts. Similarly, De Andrés-Sánchez & Puchades (2019) proposed a fuzzy-random extension of the Lee–Carter model, which integrates uncertainties in time-dependent mortality trends (using ARIMA) and age-dependent coefficients (represented by triangular fuzzy numbers). This enhanced model significantly improved the accuracy of mortality predictions, outperforming the original Lee–Carter model as well as other fuzzy-based extensions in predicting central mortality rates, death probabilities, and life expectancies.

Hong et al. (2021) expanded on this work by combining the Lee–Carter model with machine learning and statistical techniques such as random forest (RF), artificial neural networks (ANN) and ARIMA to forecast mortality rates. Their findings indicated that the LC-ANN model achieved the highest accuracy in countries with less efficient healthcare systems, such as Malaysia, whereas the LC-ARIMA model performed better in regions with more advanced healthcare infrastructures. This integration of machine learning methodologies further enhanced the precision of mortality predictions.

Raouf & Elsaieed (2022) contributed to the field by developing several fuzzy time series models, including those proposed by Chen and others, aimed at forecasting social insurance benefits. They recommended the use of the Huarng partitioning method to determine optimal interval lengths. Empirical validation confirmed that the Chen model, combined with Huarng partitioning, provided high forecasting accuracy with low error rates during both training and testing phases. This underscores the effectiveness of FTS methods in enhancing the accuracy and reliability of actuarial forecasts across diverse applications.

Fuzzy time series in other fields

On the other hand, the FTS method proved efficiency for predicting in other fields. For instance, Lee et al. (2017) improved FTS models by using the F-transform technique to reduce prediction errors, achieving greater accuracy in forecasting, such as student enrollment and Taiwanese patent data, and underscoring the value of dataset transformation. Arnita, Afnisah & Marpaung (2020) predicted rainfall in Medan using fuzzy time series models, comparing Chen (1996), Markov chain, and Cheng, Chen & Chiang (2006) variations. The Chen method proved most accurate based on the lowest MAPE values, with intervals determined using average-based principles. Alyousifi et al. (2020) used a fuzzy time series Markov chain (FTSMC) model with a grid technique to optimize partitions for predicting daily air pollution in Klang, Malaysia. This model improved prediction accuracy and outperformed traditional statistical models, making it a potentially more reliable method for forecasting air pollution. Alyousifi, Othman & Almohammedi (2021) developed a hybrid forecasting model using Markov chain and C-Means clustering for fuzzy time series data. This model showed superior performance over existing models on real datasets, including TAIEX trading data and PM10 concentrations in Melaka, Malaysia.

FTS is a broad field that comprises various models used in different fields. These models might vary based on aspects such as the choice of model, characteristics of the dataset, methodology for partitioning, and the selection of tools for assessing correctness. Therefore, the primary objective of our study is to anticipate the total benefits (pensions) within SIS in Egypt using the proposed hybrid predictive model. The primary objective of this approach is to attain exceptional forecast accuracy, as supported by empirical evidence. The primary benefit provided by the suggested model is in its ability to optimize the partitioning process, therefore mitigating computational complexity and decreasing the associated computational load in the forecasting task.

Fuzzy Time Series and Proposed Method

The concept of fuzzy set theory was first proposed by Zadeh (1965). This theory is utilized to tackle the challenges of ambiguity and uncertainty that arise in practical situations, by providing a framework for representing and analyzing linguistic fuzzy information. The utilization of FTS models has been widespread and has contributed to significant advancements in various fields. The original FTS model is developed by Song & Chissom (1993); Song & Chissom (1994) utilized the max–min composition technique, but it encountered computational difficulties. Chen’s (1996) subsequent model introduced simplifications to the calculations; however, it did not incorporate suitable weight mechanisms for fuzzy logical connections (FLRs).

Numerous researchers have since made modifications and enhancements to FTS models in order to boost the accuracy of predictions such as Chen & Hwang (2000), Huarng (2001a), Huarng (2001b), Huarng & Yu (2006), Hwang, Chen & Lee (1998) and Yu (2005). These improvements encompass a range of weighing mechanisms, extended linguistic intervals, and other division algorithms. The application of fuzzy theory, as refined by Yu (2005) has been observed in several conventional FTS models, such as those used for stock price predictions. The application of various weighted technologies was utilized by Yu to improve the accuracy of forecasting.

On the other hand, Cheng, Chen & Chiang (2006) merged FTS with a trend-weighting technique to make forecasts for both real stock prices and university enrolment. Combining fuzzy sets with the time series model generates the idea of fuzzy time series. The essential procedures for constructing fuzzy time series models encompass the establishment of a defined universe of discourse U, the partitioning of U into a uniform number of intervals, the process of fuzzification, the specification of fuzzy logic relations, the determination of forecasted values, and the subsequent defuzzification.

Previous literature has provided several key definitions of fuzzy time series that have been constructed. These definitions are outlined as follows (Chen, 1996; Song & Chissom, 1993; Song & Chissom, 1994):

Definition 1

Let X(t)(t =1 , 2, 3, .., n, ) represent a collection of real numbers, serving as the universe of discourse within which fuzzy sets fi(t) are defined. The collection F(t), denoted as f1t,f2t,…,fnt, is referred to as a fuzzy time series defined on X(t).

Definition 2

A fuzzy set refers to a category of entities that possess a continuous range of membership grades. Let U denote the Universe of discourse, which is defined as U = u1, u2, …, un, where ui represents the potential linguistic values within U. Then a fuzzy set of linguistic variables Ai of U is defined in Eq. (1) by (1) Ai=fAiu1u1+fAiu2u2+…+fAiunun,

where fAi represents the membership function of fuzzy set Ai, and fAiu1 represents the degree of belonging of u1 to Ai, if ui is a member of Ai.

Definition 3

The fuzzy logic relationship (FLR) between F(t − 1) and F(t), indicated as R(t, t − 1), can be represented as F(t − 1) → F(t). If R(t, t − 1) is independent of t for any given t value, then, (2) Rt,t−1=Rt−1,t−2.

Definition 4

Let us assume that the value of F at time t−1 is denoted as Ai, and the value of F at time (t) isdenoted as Aj. The FLR, or Forward-Looking Relationship, is a term used to describe the connection between two consecutive observations, F(t − 1) and F(t). It is denoted as Ai → Aj, where Ai represents the left-hand side of the FLR and Aj represents the right-hand side.

Tree partition method (TPM)

TPM methodology is a partition approach introduced by Alyousifi, Othman & Almohammedi (2021) to linguistic partitioning that has been devised as a way for re-partitioning, utilizing the average length of the initial sub-intervals, particularly the average inter-quartile range. The method outlined in this study is characterized by its simplicity and effectiveness in determining the optimal interval lengths for obtaining an ideal partition of the universe of discourse (U), in contrast to clustering methods. In the majority of research conducted on fuzzy time series, U is partitioned into intervals of equal duration. However, fuzzy time series models may not provide aesthetically pleasant prediction outcomes in situations where the distribution of the universe of speech is not uniform. Therefore, the tree partition method is implemented in this study. Figure 1 shows the flowchart of TPM, and the steps of the suggested partition method are outlined in detail as follows:

Figure 1 The flowchart of TPM algorithm.

First step: The universe of discourse, denoted as U, is defined.

Second step: This universe is partitioned into a minimum of three and a maximum of five equal-length intervals.

Third step: Calculate the frequency of each sub-interval based on the observations in data. Then, the average of sub-intervals’ frequencies is calculated.

Fourth step: To optimize the granularity of the intervals, if any of the sub-intervals exceed the computed average length, a further division of the specific sub-interval into two equal parts is undertaken. This procedure is repeated until no individual sub-interval is longer than the average length at the outset.

Fifth step: In the end, the number of partitions where the smallest sub-interval is shorter than the average is taken into account for subsequent calculations and analysis.

The proposed model

In this study, a hybrid predictive model that integrates fuzzy time series Markov chain (FTSMC) with the TPM and the difference transformation data method for predicting the total benefits (pensions) of the SIS in Egypt. The proposed framework utilizes the simplified arithmetic operations of the FTSMC model, as developed by Tsaur (2012). The phases of the suggested model are illustrated in Fig. 2 and can be delineated in detail as follows:

Figure 2 The proposed model’s framework.

Step 1:

This step collects the total benefits (pensions) of SIS in Egypt and preprocessing the data using the difference transformation data method and splitting data into training and testing datasets using the holdout (80:20) splitting approach.

Step 2:

The universe of discourse (U) is defined and established based on the dataset collection using Eq. (3) below: (3) U=Dmin−D1,Dmax+D2,

where the minimum number of observations is Dmin andthe maximum is Dmax. The numbers D1 and D2 are both positive integers.

Step 3:

In this step, the universe of discourse (U) is divided into numerous partitions, and the optimal number of partitions is determined using the TPM algorithm, as discussed in subsection (tree partition method).

Step 4:

The fuzzy set, denoted as Ai, is a collection of things that possess a continuous membership grade. This membership grade is determined using the universe of discourse (U), which is defined as u1, u2, …, un, as specified in Eq. (1). Next, the fuzzy sets are established for each real value of the time series by assigning them fuzzy numbers determined by the maximum membership value, following the intervals outlined in Step 3.

Step 5:

In this step, the fuzzy logical relationships (FLRs) are established between the linguistic values Ft−1=Ai and Ft=Aj, denoted as Ai → Aj, for all FLRs. For instance, FLRs such as A1 → A1, A1 → A2, and A1 → A3 are determined. Subsequently, the fuzzy logical relation groups (FLRGs) are categorized and transformed into groups A1 → A1, A2 and A3.

Step 6:

The Markov weights, also known as the transition probability matrix, are derived from the frequencies of the established FLRGs in Step 6. The matrix P is Pn×n. The state transition probability, denoted as Pij, is the probability of transitioning from state Ai to state Aj. This probability can be determined using Eq. (4) as follows: (4) Pij=NijNi,i,j=1,2,3,…..,n,

where Nij represents the count of transitions from state Ai to state Aj, while Ni represents the overall count of transitions in state Ai.

Step 7:

Determine the forecasted values through the process of forecasting. The calculation of forecasts considers the following rules. Two cases are being considered: one-to-one and one-to-many.

Rule 1, when the fuzzy logical relationship group of Ai exhibits a one-to-one relationship, meaning there is only one transition for Ai, the forecasting of F(t) is determined by mk, which represents the midpoint of uk. Here, k ranges from 1 to n. This calculation may be performed using the Eq. (5) as follows: (5) Ft+1=mk.Pij=mk.

Rule 2. when Ai fuzzy logical relationship group contains more than one transition, also known as “many-to-one”. To calculate the predicted value F(t + 1), the following Eq. (6) can be used if the state is Ai for the actual value Y(t) at time t. (6) Ft+1=m1.Pi1+m2.Pi2+…+Yt.Pii+…+mk.Pin,

where m1, …., mk represents the midpoint of u1, …., uk.

Step 8:

Calculate the final value of prediction after adding adjusted values by using Eq. (7) as shown below: (7) F∗t+1=Ft+1±D,

where F∗trepresents the final prediction value after the adjustment, Ft isthe prediction value before the adjustment. D is the adjustment value according to the differences to reduce the estimated error.

Data & Evaluation Methods

Data description and preprocessing of pension payments in Egypt

The dataset presented in this study was acquired from authoritative sources, namely the National Fund for Social Security and the Ministry of Insurance and Social Affairs (MOISA) (https://www.capmas.gov.eg/Pages/Publications.aspx?page_id=5104&Year=23578). It encompasses data from a substantial 44-year period, from 1976 to 2019, with annual statistics specifically drawn from chapter/section 17 titled “Social Care” during the period 1979–2019. The records apply to the total benefits that are required by the social insurance system, which include contributions from the government sector, public and private sector. Table 1 and Fig. 3 show the total benefits of Egyptian social insurance system in million during this study period.

Table 1 The dataset description.

Year	Private sector	Government sector	Total benefits in millions	Year	Private sector	Government sector	Total benefits in millions	
1976	58	121	179	1998	5003	5252	10255	
1977	40	145	185	1999	5846	6034	11880	
1978	52	162	214	2000	6798	6689	13487	
1979	69	195	264	2001	7563	7742	15305	
1980	105	243	348	2002	8314	9590	17904	
1981	121	207	328	2003	8996	10791	19787	
1982	352	401	753	2004	9756	12240	21996	
1983	425	436	861	2005	10590	13988	24578	
1984	485	557	1042	2006	12865	15441	28306	
1985	553	641	1194	2007	13398	16867	30265	
1986	631	779	1410	2008	15170	19311	34481	
1987	688	841	1529	2009	18139	19488	37627	
1988	824	1013	1837	2010	18456	22660	41116	
1989	969	1275	2244	2011	22124	28724	50848	
1990	1134	1417	2551	2012	28164	35568	63732	
1991	1386	1714	3100	2013	35777	33640	69417	
1992	1706	2037	3743	2014	43175	40558	83733	
1993	2085	2353	4438	2015	51816	48554	100370	
1994	2563	3019	5582	2016	59169	55495	114664	
1995	3000	3587	6587	2017	67600	62300	129900	
1996	3470	4107	7577	2018	77600	72800	150400	
1997	4091	4810	8901	2019	92800	85700	178500	

To preprocess the data, we utilize the first difference transformation technique and select the holdout (80:20) splitting approach. The division of data into training and testing sets is a fundamental and essential practice in statistical modelling. This process plays a critical role in evaluating the performance of the model and assuring its ability to effectively generalize to new, unseen data. By training the model on one subset of the data and then evaluating it on a distinct subset that it hasn’t seen before (Joseph, 2022; Kahloot & Ekler, 2021). Based on the outcomes of our extensive review of the academic literature, it has been observed that the holdout (80:20) splitting methodology is a widely employed method for data partitioning, as previously employed in various studies (Hasanov, Wolter & Glende, 2022; Joseph & Vakayil, 2022), to divide our time series data into two independent sets. The initial dataset comprised 80% of 2009 and covering the time frame from 1976 to 2009 and was employed for the development of the models deployed in the training process. The second subset consisted of 20% of the data collected between 2010 and 2019. This subset was utilized to validate and assess the effectiveness of the proposed model.

Figure 3 The total benefit (in millions) graph.

In this study, we chose to model pension benefits using an additive approach, rather than the commonly used multiplicative model, for several key reasons. First, the historical data we analyzed demonstrated non-linear and often irregular growth patterns, particularly during periods of economic instability and policy changes. These variations were not adequately captured by multiplicative models, which tend to assume more consistent exponential growth. The additive model, on the other hand, allowed us to represent incremental changes more accurately, particularly during periods of slower or more erratic growth, where the assumption of constant percentage increases would lead to overestimations. Additionally, the additive approach provides greater flexibility in scenarios where pension increments are driven by fixed monetary adjustments rather than proportional increases, which was observed in some segments of our data. By employing an additive model, we aimed to reflect the real-world dynamics of the pension system under study, acknowledging that while multiplicative models may be more suitable for scenarios with consistent growth, the additive model better fits the specific characteristics of our dataset.

The evaluation metrics of forecasting models

The evaluation of a model’s performance is dependent on its ability to generate forecast values that are closely aligned with the observed values for testing dataset. To evaluate the efficacy of the proposed model, four distinct forecast consistency measures were utilized. Four statistical criteria, namely mean absolute percent error (MAPE), Thiels’ U statistic, and root mean square error (RMSE), were adopted as measures of forecasting accuracy to assess the models (Khalil, Liu & Ali, 2024b; Khalil et al., 2022b; Rashidpoor Toochaei & Moeini, 2023), as defined in Eqs. (8)–(10), respectively. The measures are presented as follows:

(8) MAPE=100n∑t=1nyt−yt∗yt,

(9) Thiels′U=∑t=1nyt−yt∗2∑t=1nyt2+∑t=1nyt∗2,

(10) RMSE=1n∑t=1nyt−yt∗2,

where yt and yt∗ denote actual and predicted model data samples, respectively. n is the sample size. y¯t is the average of actual data.

Evaluation metrics for the prediction interval of the proposed model

Prediction intervals are typically assessed by their ability to accurately encompass the target values (reliability) and their overall width (sharpness). Two widely used, independent metrics for evaluating precise prediction intervals are the prediction interval coverage probability (PICP), which measures reliability, and the mean prediction interval length (MPIL), which measures sharpness. These metrics are briefly described below.

Prediction interval coverage probability (PICP)

PICP measures the proportion of observed data points that fall within the prediction intervals generated by a model. It indicates how well the prediction intervals capture the true values of the target variable (Adjenughwure & Papadopoulos, 2020). The PICP is defined by using Eqs. (11) and (12) as follows: (11) PICP= ∑t=1nCtn,

where, (12) Ct=1ifytL≤yt≤ytU0otherwise,

where Ct refers to the number of observations within the prediction intervals, and n refers to the total number of observations.

Mean prediction interval length (MPIL)

MPIL quantifies the average width of the prediction intervals for all observations. It reflects the precision of the predictions, with narrower intervals indicating greater prediction accuracy (Tsao, Leu & Chou, 2021). The MPIL is defined by using Eq. (13) as follows: (13) MPIL=1n∑t=1nytU−ytL,

where ytU and ytL are the upper and lower prediction limits for yt.

Results and Discussion

This section provides a comprehensive calculation of the forecasting methods used in the study, including the FTS and statistical employed. The aim is to provide a clear account of the methodologies employed for data analysis, ensuring transparency and precision.

Experimental setup

The experiments were conducted on a computer equipped with a 2.60 GHz Intel(R) Core (TM) i7-12700F CPU and 32 GB of RAM, running a 64-bit version of Windows 11. The framework was implemented using Python, with the dataset loaded via the Pandas data frame. FTS methods were executed using the PYFTS library. The source code and data associated with the proposed work have been made publicly accessible on the author’s GitHub page (https://github.com/AhmedKhalil91/The-Benefits-of-Social-Insurance-System-prediction.git).

The proposed model application

The calculations and implementation of the suggested method are fully illustrated in this subsection. The procedure is as follows:

(1) Define the universe of discourse (U) from SIS data using Eq. (3).

U=Dmin−D1,Dmax+D2

U=−20−5,4216+4

U=−25,4220.

(2) Divide the universe of discourse U based on TMP algorithm. Table 1 shows the implementation of the proposed partition method based on the step that is illustrated in the methodology section. Consequently, it has been determined that the frequencies of u1, andu2 are greater than the mean. Subsequently, the intervals are divided into halves. Then, we checked that again until we found there was not any sub-interval larger than the average. According to the result of TPM implementation, the final number of partitions of U is nine sub-intervals as shown in Table 2.

Table 2 TPM training dataset procedures description.

1st partition	2nd partition	3rd partition	4th partition	
Interval	Frequency	Interval	Frequency	Interval	Frequency	Interval	Frequency	
[−25, 824]	18	[−25, 339.5]	13	[−25, 187.25]	10	[−25, 81.125]	5	
[824, 1673]	7	[399.5, 824]	5	[187.25, 399.5]	3	[81.125, 187.25]	5	
[1673, 2522]	4	[824, 1248.5]	3	[399.5, 824]	5	[187.25, 399.5]	3	
[2522, 3371]	3	[1248.5, 1673]	4	[824, 1248.5]	3	[399.5, 824]	5	
[3371, 4220]	2	[1673, 2522]	4	[1248.5, 1673]	4	[824, 1248.5]	3	
		[2522, 3371]	3	[1673, 2522]	4	[1248.5, 1673]	4	
		[3371, 4220]	2	[2522, 3371]	3	[1673, 2522]	4	
				[3371, 4220]	2	[2522, 3371]	3	
						[3371, 4220]	2	
Repartition	>6.8	Repartition	>6.8	Repartition	>6.8	Repartition	>6.8	
The average of frequency for the first partition = 18+7+4+3+25=6.8	
If the frequency of all sub intervals is less than 6.8, then the partitioning process will end. Otherwise, the process of repartition should be continued	
Notes.

The bold numbers represent the interval frequency that exceeds the average frequency of all sub-intervals.

(3) Convert each time series’ real value into a fuzzy number using the intervals you just put up as a guide, and then define the corresponding fuzzy sets using the maximum membership value. Table 3 shows the fuzzy sets with the mid-point of each interval and its fuzzy numbers. The linguistic time series of maximum membership is shown in Table 4.

Table 3 The fuzzy-numbered intervals.

Number of intervals	Interval u i	Mid-point of interval ( m i )	Interval code	Fuzzy number	
1	[−25, 81.125]	28.0625	u 1	A 1	
2	[81.125, 187.25]	134.1875	u 2	A 2	
3	[187.25, 399.5]	293.375	u 3	A 3	
4	[399.5, 824]	611.75	u 4	A 4	
5	[824, 1248.5]	1036.25	u 5	A 5	
6	[1248.5, 1673]	1460.75	u 6	A 6	
7	[1673, 2522]	2097.5	u 7	A 7	
8	[2522, 3371]	2946.5	u 8	A 8	
9	[3371, 4220]	3795.5	u 9	A 9	

(4) The fuzzy logic relationships (FLRs) are established based on the training dataset, for instance, FLRs such as A1 → A1, A1 → A2, as shown in Table 5. Subsequently, the fuzzy logical relation groups (FLRGs) are categorized and transformed into groups A1 → A1, A2 and A3, as shown in Table 6.

Table 4 Linguistic time series.

No	Linguistic time series values A i	
1	A1= 1u1+0.5u2+0u3+0u4+0u5+0u6+0u7+0u8+0u9	
2	A2= 0.5u1+1u2+0.5u3+0u4+0u5+0u6+0u7+0u8+0u9	
3	A3= 0u1+0.5u2+1u3+0.5u4+0u5+0u6+0u7+0u8+0u9	
4	A4= 0u1+0u2+0.5u3+1u4+0.5u5+0u6+0u7+0u8+0u9	
5	A5= 0u1+0u2+0u3+0.5u4+1u5+0.5u6+0u7+0u8+0u9	
6	A6= 0u1+0u2+0u3+0u4+0.5u5+1u6+0.5u7+0u8+0u9	
7	A7= 0u1+0u2+0u3+0u4+0u5+0.5u6+1u7+0.5u8+0u9	
8	A8= 0u1+0u2+0u3+0u4+0u5+0u6+0.5u7+1u8+0.5u9	
9	A9= 0u1+0u2+0u3+0u4+0u5+0u6+0u7+0.5u8+1u9	

Table 5 Dataset training fuzzy number and fuzzy logic relationships.

Year	Total benefits in millions	Difference	Fuzzy number	Fuzzy logic relationships	
1976	179	0	A 1	–	
1977	185	6	A 1	A1 → A1	
1978	214	29	A 1	A1 → A1	
1979	264	50	A 1	A1 → A1	
1980	348	84	A 2	A1 → A2	
1981	328	-20	A 1	A2 → A1	
1982	753	425	A 4	A1 → A4	
1983	861	108	A 2	A4 → A2	
1984	1042	181	A 2	A2 → A2	
1985	1194	152	A 2	A2 → A2	
1986	1410	216	A 3	A2 → A3	
1987	1529	119	A 2	A3 → A2	
1988	1837	308	A 3	A2 → A3	
1989	2244	407	A 4	A3 → A4	
1990	2551	307	A 3	A4 → A3	
1991	3100	549	A 4	A3 → A4	
1992	3743	643	A 4	A4 → A4	
1993	4438	695	A 4	A4 → A4	
1994	5582	1144	A 5	A4 → A5	
1995	6587	1005	A 5	A5 → A5	
1996	7577	990	A 5	A5 → A5	
1997	8901	1324	A 6	A5 → A6	
1998	10255	1354	A 6	A6 → A6	
1999	11880	1625	A 6	A6 → A6	
2000	13487	1607	A 6	A6 → A6	
2001	15305	1818	A 7	A6 → A7	
2002	17904	2599	A 8	A7 → A8	
2003	19787	1818	A 7	A8 → A7	
2004	21996	1818	A 7	A7 → A7	
2005	24578	2599	A 8	A7 → A8	
2006	28306	3728	A 9	A8 → A9	
2007	30265	1959	A 7	A9 → A7	
2008	34481	4216	A 9	A7 → A9	
2009	37627	3146	A 8	A9 → A8	

Table 6 Fuzzy logic relationships groups (FLRG) for training dataset.

No	FLRG	
G1	A1→3A1,1A2,1A4	
G2	A2→1A1,2A2,2A3	
G3	A3→1A2,2A4	
G4	A4→1A2,1A3,2A4,1A5	
G5	A5→2A5,1A6	
G6	A6→3A6,1A7	
G7	A7→1A7,2A8,1A9	
G8	A8→1A7,1A9	
G9	A9→1A7,1A8	

(5) The transition probability matrix is calculated based on FLRGs that were created in the previous step. Table 7 shows the Markov transition probability matrix based on the probability of transitioning calculated by the frequencies of the established FLRGs using Eq. (4).

Table 7 Markov transition probability matrix and count matrix based on FLRGs.

	1	2	3	4	5	6	7	8	9	
1	F	3	1	0	1	0	0	0	0	0	
P	0.6	0.2	0	0.2	0.0	0.0	0.0	0.0	0.0	
2	F	1	2	2	0	0	0	0	0	0	
P	0.2	0.4	0.4	0.0	0.0	0.0	0.0	0.0	0.0	
3	F	0	1	0	2	0	0	0	0	0	
P	0.0	0.33	0.0	0.67	0.0	0.0	0.0	0.0	0.0	
4	F	0	1	1	2	1	0	0	0	0	
P	0.0	0.2	0.2	0.4	0.2	0.0	0.0	0.0	0.0	
5	F	0	0	0	0	2	1	0	0	0	
P	0.0	0.0	0.0	0.0	0.67	0.33	0.0	0.0	0.0	
6	F	0	0	0	0	0	3	1	0	0	
P	0.0	0.0	0.0	0.0	0.0	0.75	0.25	0.0	0.0	
7	F	0	0	0	0	0	0	1	2	1	
P	0.0	0.0	0.0	0.0	0.0	0.0	0.25	0.5	0.25	
8	F	0	0	0	0	0	0	1	0	1	
P	0.0	0.0	0.0	0.0	0.0	0.0	0.5	0.0	0.5	
9	F	0	0	0	0	0	0	1	1	0	
P	0.0	0.0	0.0	0.0	0.0	0.0	0.5	0.5	0.0	
Notes.

The bold numbers indicate the Markov transition probability for each interval based on FLRGs.

(6) Calculate the forecasted values by using the rules that are illustrated in the proposed methodology, with Eqs. (5) and (6) based on Markov transition probability matrix. For example, the forecast value for year (1978) is calculated as following:

F1978=Yt.p11+m2.p12+m4.p14

=6∗0.6+134.187∗0.2+611.75∗0.2=152.78

(7) Utilize Eq. (7) to compute the final predicted values. For instance, the subsequent formula is utilized to compute the final forecast value for total benefits in the year 1978: F∗1978=152.78+23=175.78

Then, the forecasted total benefits 1978=175.78+185=360.78

Table 8 illustrates the forecasted values for the total benefits of Egyptian social insurance system in million. The fitted value of the proposed model for training and testing datasets is shown in Fig. 4.

Table 8 Dataset training fuzzy number and fuzzy logic relationships.

Year	Total benefits	Difference	Forecast value	Adjusted value	Adjusted Forecast value	Forecasted Total benefits	
1976	179	0	–	–	–	–	
1977	185	6	149.18	6	149.18 + 6 = 155.18	155.18 + 179 = 334.18	
1978	214	29	152.78	23	175.78	360.78	
1979	264	50	166.58	21	187.58	401.58	
1980	348	84	142.96	34	176.96	440.96	
1981	328	−20	199.58	−104	95.58	443.58	
1982	753	425	104.76	445	549.76	877.76	
1983	861	108	292.96	−317	−24.03	728.96	
1984	1042	181	166.16	73	239.16	1100.16	
1985	1194	152	195.36	−29	166.36	1208.36	
1986	1410	216	454.15	64	518.15	1712.15	
1987	1529	119	209.36	−97	112.36	1522.36	
1988	1837	308	454.15	189	643.15	2172.15	
1989	2244	407	415.96	99	514.96	2351.96	
1990	2551	307	454.15	-100	354.15	2598.15	
1991	3100	549	415.56	242	657.56	3208.56	
1992	3743	643	512.36	94	606.36	3706.36	
1993	4438	695	549.96	52	601.96	4344.96	
1994	5582	1144	947.69	449	1396.69	5834.69	
1995	6587	1005	1248.50	−139	1106.53	6688.53	
1996	7577	990	1155.40	−15	1140.40	7727.40	
1997	8901	1324	1266.88	334	1600.88	9177.88	
1998	10255	1354	1517.38	30	1547.38	10448.38	
1999	11880	1625	1539.88	271	1810.88	12065.88	
2000	13487	1607	1743.13	−18	1725.13	13605.13	
2001	15305	1818	2823.88	211	3034.88	16521.88	
2002	17904	2599	2946.50	781	3727.50	19032.50	
2003	19787	1818	3071.88	−781	2290.88	20194.88	
2004	21996	1818	2876.63	0	2876.63	22663.63	
2005	24578	2599	2946.50	781	3727.50	25723.50	
2006	28306	3728	2522	1129	3651	28229	
2007	30265	1959	3354.13	−1769	1585.13	29891.13	
2008	34481	4216	2522	2257	4779	35044	
2009	37627	3146	2946.50	−1070	1876.50	36357.50	

The proposed model’s evaluation for the prediction interval

The proposed model’s outputs are evaluated for training and testing data using the established metrics of Prediction Interval Coverage Probability (PICP) and mean prediction interval length (MPIL) to assess the quality of the prediction intervals and compared to other traditional fuzzy and statistical time series models. A 95% prediction interval is used as the baseline, with fuzzy numbers representing all intervals from 95% down to 0%. As presented in Table 9, the PICP is approximately 95%, suggesting that the prediction intervals are well-calibrated with near-perfect coverage. However, the MPIL of 424.5 indicates that the model generates relatively wide intervals, reflecting a substantial degree of uncertainty in the data. This result suggests that the proposed model effectively balances coverage accuracy and interval precision.

The results of the proposed model’s evaluation

In this subsection, the proposed model’s performance is evaluated using three statistical criteria, namely mean absolute percent error (MAPE), Thiels’ U statistic, and root mean square error (RMSE) for the training and testing datasets. In addition, to validate the proposed model, we introduce a comparison of the proposed model with existing traditional fuzzy time series models that were proposed by Song & Chissom (1994), Chen (1996), Yu (2005), and Cheng Chen & Chiang (2006). It is important to note that all the fuzzy time series models in the comparison, including the proposed model, were evaluated based on the same number of intervals; specifically, nine intervals. This uniformity ensures that the comparative results are fair and allows for a direct assessment of the relative performance of the models under the same interval partitioning structure.

Figure 4 Plots of fit proposed model for training and testing datasets with the actual values of total benefits.

Table 9 Evaluating model prediction intervals.

	PICP	MPIL	
Linear regression	42.86%	1119.01	
ARIMA	60%	2250.63	
Exponential Smooth	60%	2250.63	
Song & Chissom (1994)	100%	471.67	
Chen (1996)	100%	471.67	
Yu (2005)	100%	471.67	
Cheng, Chen & Chiang (2006)	100%	471.67	
The proposed model	100%	424.5	

Table 10 reveals that the proposed model had the best overall performance in predicting the total benefits of social insurance system in Egypt, compared to other traditional fuzzy time series models, as it had lower values of MAPE, RMSE, and Thiels’ U for training dataset with values of 11.8, 472.67, and 0.016 respectively. As well as, for testing dataset with values of 10.65, 12999.6, and 0.064 respectively, as shown in Fig. 5.

Table 10 The evaluation of the proposed model with traditional fuzzy time series models.

Model	Evaluation metric	Rank	
	Training dataset	Testing dataset		
	RMSE	MAPE%	Thiels’ U	RMSE	MAPE%	Thiels’ U		
Song & Chissom (1994)	879.41	17.31	0.033	13314.2	11.25	0.069	5th	
Chen (1996)	553.26	16.76	0.025	13306.6	11.05	0.068	4th	
Yu (2005)	501.50	12.92	0.02	13305.5	10.98	0.067	2nd	
Cheng, Chen & Chiang (2006)	504.14	13.24	0.023	13306.6	11.05	0.068	3rd	
The proposed model	472.67	11.80	0.016	12999.6	10.65	0.064	1st	
Notes.

The bold numbers denote the lowest values of the evaluation metrics achieved by the proposed model.

Figure 5 Plots of the evaluation metrics for training and testing datasets for models.

The evaluation results indicate that the proposed model significantly outperforms conventional models like linear regression, ARIMA, and exponential smoothing in both training and testing datasets as shown in Table 11. With the lowest RMSE (472.67 for training and 12,999.64 for testing), MAPE (10.653% for testing), and Thiels’ U (0.064 for testing), the proposed model demonstrates superior accuracy and generalization ability. In contrast, linear regression exhibits the poorest performance, particularly in the testing set, with an RMSE of 78,080.63. While ARIMA and exponential smoothing show comparable results, their errors are notably higher than the proposed model’s, highlighting the proposed model’s reliability and predictive efficiency.

Table 11 The evaluation of the proposed model with conventional statistical models.

Model	Evaluation metric	
	Training dataset	Testing dataset	
	RMSE	MAPE%	Thiels’ U	RMSE	MAPE%	Thiels’ U	
Linear Regression	4745.73	452.78	0.169	78080.63	62.82	0.564	
ARIMA	805.08	12.03	0.028	52078.54	34.13	0.314	
Exponential Smooth	551.10	13.86	0.019	52503.97	34.46	0.317	
The proposed model	472.67	11.80	0.016	12999.64	10.653	0.064	
Notes.

The bold numbers also represent the lowest values of the evaluation metrics achieved by the proposed model.

Furthermore, Table 11 emphasizes the broader implications of this study for the insurance sector, particularly in enhancing operational efficiency through an evaluation of Egypt’s social insurance system and the development of precise analytical and predictive methods. The study’s focus on the FTS model, known for its ability to capture complex, non-linear patterns and uncertainties, further highlights its effectiveness in the Egyptian insurance market, where conventional statistical approaches often face challenges due to data constraints. The insights gained from this research provide valuable tools for insurers aiming to improve risk management and profitability, while also holding relevance for stakeholders such as regulators and consumers, potentially leading to more accurate insurance pricing and improved policy frameworks.

Conclusions

In conclusion, enhancing the accuracy of predicting social insurance benefits in Egypt has far-reaching implications for countries around the world. Social insurance systems are vital components of social welfare and economic stability, serving as a safety net for individuals and providing economic security during times of need. Therefore, improving prediction accuracy in this sector is essential for the efficient allocation of resources and the sustainability of these systems. Accurate forecasting enables policymakers and insurance providers to anticipate future needs, allocate resources more effectively, and adjust policies proactively, ensuring that the system remains robust and responsive to changing social and economic conditions.

Integrating actuarial methods with predictive analytics enhances understanding and anticipation of future trends, supporting better decision-making and strengthening social insurance systems worldwide. Recently, hybrid predictive methods have garnered significant attention. This research introduces a novel hybrid predictive model combining the fuzzy time series Markov chain, TPM, and the difference transformation data method to forecast the total benefits (pensions) within Egypt’s social insurance system. The datasets for this study were sourced from the National Fund for Social Security and the Ministry of Insurance and Social Affairs (MOISA). The main goal of this model is to achieve high forecast accuracy, as demonstrated by empirical evidence. The proposed model’s key advantage lies in optimizing the partitioning process, thereby reducing computational complexity and load during forecasting.

The model’s performance was assessed using standard statistical evaluation metrics such as RMSE, MAPE, and Thiel’s U statistic, on both training and testing datasets. The results were compared with traditional fuzzy time series models proposed by Song & Chissom (1994), Chen (1996), Yu (2005), and Cheng, Chen & Chiang (2006). Findings indicated that the proposed hybrid model significantly improved forecasting accuracy by minimizing randomness and variability in the data. The model proved to be more robust and outperformed traditional fuzzy time series approaches. This study contributes to the literature by presenting a mathematical model applicable to estimating crucial values in the insurance sector. It also opens up several avenues for future research on enhancing predictive accuracy in social insurance systems.

The main limitations of our study include the use of a first-order FTS model, which, while simpler and less prone to overfitting, may limit the complexity of the predictions. Additionally, the dataset was constrained to 44 years of data, focusing exclusively on the Egyptian market, which may restrict the generalizability of the findings. The interpretability of the fuzzy rules is also a potential limitation, as understanding the underlying reasons for the model’s predictions could be improved.

Future research should explore alternative partition methods and higher-order or hybrid FTS models, potentially integrating techniques such as ARIMA or exponential smoothing, to enhance the interpretability and robustness of the model. Applying these methods to datasets from different countries would also help increase the model’s generalizability. Furthermore, the approach presented in this study could be extended to predict other critical ratios within the insurance sector, potentially offering strong performance in forecasting time series data in this field.

Additional Information and Declarations

Competing Interests

Author Contributions

Data Availability

The authors declare there are no competing interests.

Ahmed Abdelreheem Khalil conceived and designed the experiments, performed the experiments, analyzed the data, performed the computation work, prepared figures and/or tables, authored or reviewed drafts of the article, and approved the final draft.

Mohamed Abdelaziz Mandour conceived and designed the experiments, analyzed the data, prepared figures and/or tables, and approved the final draft.

Ahmed Ali performed the experiments, performed the computation work, authored or reviewed drafts of the article, and approved the final draft.

The following information was supplied regarding data availability:

The raw data and code are available in the Supplementary File.

The dataset is available at GitHub and Zenodo:

- https://github.com/AhmedKhalil91/The-Benefits-of-Social-Insurance-System-prediction.git

- Khalil, A. (2024). The Benefits of Social Insurance System prediction using a hybrid Fuzzy Time Series method. Zenodo. https://doi.org/10.5281/zenodo.13987690.

The dataset used in the research was acquired from the National Fund for Social Security and the Ministry of Insurance and Social Affairs (MOISA): https://www.capmas.gov.eg/Pages/Publications.aspx?page_id=5104&Year=23578.

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
