# Peer review of "The benefits of social insurance system prediction using a hybrid fuzzy time series method"

_PeerJ Computer Science, doi:10.7717/peerj-cs.2500_

## Round 0.1 · original submission · Major Revisions

· Academic Editor

Major Revisions

Dear authors,

Thank you for submitting your article. Reviewers have now commented on your article and suggest major revisions. We do encourage you to address the concerns and criticisms of the reviewers and resubmit your article once you have updated it accordingly. By the way, some paragraphs are too long to read. They should be divided into two or more for readability and comprehensibility. Equations should be used with equation number. Explanation of the equations should be checked. Definitions and boundaries of all variables should be provided. Necessary references should also be given.

Best wishes,

Reviewer 1 ·

Basic reporting

The authors propose a modeling of fuzzy time series based on the proposals of the works by Song and Chisom, applied to the prediction of pension payment increases. Although the proposal could potentially be of interest, it presents several drawbacks that need to be addressed. The order of the queries does not imply importance.
1. The work needs a review in terms of format, typos, etc. The way of citing works does not adhere to an academic format (e.g., APA), there are numerous misplaced commas and periods, and there are expressions like "language mathematical reasoning" that make no sense, etc.
2. Some statements that motivate the work need to be supported by citations, for example, line 72 and following, and line 111 and following.
3. The review of the works on lines 176 and following in a single paragraph makes it practically unreadable. It should be better structured by themes and instruments. Additionally, the use of fuzzy techniques in strictly insurance issues (such as, e.g. predicting mortality) is not addressed.
4. The review of the fuzzy time series methods on lines 227 and following presents the same problem as on lines 176 and ff: it is done in a single, very long, and practically unreadable paragraph.
5. The authors do not report how retirement benefits have been predicted in the purely actuarial literature. The approach used (predicting benefits from the previous ones instead of predicting the population's mortality) seems very basic and should be referenced from the actuarial literature perspective.
6. The incremental benefits are modeled in a summative way instead of a multiplicative one when usually this type of magnitude is multiplicative, that is, pensions grow by X% annually, not by 200 monetary units annually. As time goes by, the lower summative intervals are less likely to occur as the volume of benefits has increased.
7. It is necessary to discuss the approach used in comparison to those currently existing (whether fuzzy or not) and to highlight the potential theoretical and practical implications of the proposed fuzzy time series model.

Experimental design

Data and procedure is clearly presented. Issues come in how the method predicts (point estimates instead confidence intervals). See above comments.

Validity of the findings

The authors model the predictions as fuzzy subsets. However, ultimately, the authors only compare the accuracy of the models based on an expected value that arises from the confidence intervals that quantify the possible variations in pensions. The interest in using fuzzy subsets is that predictions can be expressed as confidence intervals without significant computational cost. In this sense, I recommend:
- If the authors intend to make point predictions, they should compare their method with simpler/traditional ones like linear time trending or ARMA.
- It would be desirable for the authors to demonstrate the ability to predict using confidence intervals of the magnitude of interest. In this sense, appropriate metrics could be the Prediction Interval Coverage Probability or the Mean Prediction Interval Length to compare between methodologies that predict with confidence intervals.

Reviewer 2 ·

Basic reporting

The study adds to the growing literature on the Social Insurance System (SIS) in Egypt by predicting the benefits of SIS. The study reveals several implications for theory and practice related to the key benefits of SIS. Findings could help governments create a precise Social Insurance System with less misuse and budget. The manuscript is well-developed and presents a relevant and satisfactory theoretical framework.

Experimental design

Although some sections present little updated (or old) content, the manuscript presents (overall) current and innovative content. We would like the manuscript to show a greater real contribution (theoretical, practical, and scientific) at the end of the research.

Validity of the findings

If the authors provide few managerial implications to the industry based on this study's key findings in section 4.

Additional comments

Some tables present relevant content but with little reflection and critical sense. We suggest reinforcing the limitations, conclusions, and future research. We suggest that English be revised (minor revisions) for greater publication potential.

---

## Round 0.2 · Major Revisions

· Academic Editor

Major Revisions

Dear authors,

Thank you for the revision. One of the original reviewers did not respond to the invitation for reviewing your revised manuscript. According to Reviewer1, your paper still needs a revision and we encourage you to address the concerns and criticisms of this reviewer and resubmit your article once you have updated it accordingly.

Best wishes,

Reviewer 1 ·

Basic reporting

The text presented by the authors has improved in some aspects. However, in others, nothing has changed, and in some cases, it is very difficult to notice the changes since there is no yellow highlighted text, contrary to what the authors indicated. Thus, I point out that, based on the initial review, the authors have not addressed the following points:
Point 1. The work needs a review in terms of format, typos, etc. The way of citing works does not adhere to an academic format (e.g., APA), there are numerous misplaced commas and periods, and there are expressions like "language mathematical reasoning" that make no sense, etc.
Not addressed. For example, the citation in line 70 does not follow APA style. The same title, from a strictly linguistic point of view, should be improved. These are just two examples.
Point 3. The review of the works on lines 176 and following in a single paragraph makes it practically unreadable. It should be better structured by themes and instruments. Additionally, the use of fuzzy techniques in strictly insurance issues (such as, e.g. predicting mortality) is not addressed.
Point 4. The review of the fuzzy time series methods on lines 227 and following presents the same problem as on lines 176 and ff: it is done in a single, very long, and practically unreadable paragraph.
Regarding points 3 and 4, the presentation has improved, but it is still of complicate reading: works on fuzzy time series with completely different approaches are cited together without any structure. Explanations about two prediction works using fuzzy in the actuarial field are extended within the same paragraph. The topic of "fuzzy time series" and "fuzzy time series in actuarial science" is included without any clear structure. These should be developed in a more structured manner, slightly more detailed, and with two distinct subheadings.
Point 5. The authors do not report how retirement benefits have been predicted in the purely actuarial literature. The approach used (predicting benefits from the previous ones instead of predicting the population's mortality) seems very basic and should be referenced from the actuarial literature perspective.
This aspect has either not been implemented or, as presented, it is very difficult to verify. How has the pension increase been modeled in the actuarial studies? It is not developed..
Point 6. The incremental benefits are modeled in a summative way instead of a multiplicative one when usually this type of magnitude is multiplicative, that is, pensions grow by X% annually, not by 200 monetary units annually. As time goes by, the lower summative intervals are less likely to occur as the volume of benefits has increased.
By observing the time series that you provide, it is clear that the first difference between two observations is not “constant.” Furthermore, to the best of my knowledge, pension fluctuations are always modeled in a multiplicative way. It should be better justified why the authors model them additively, as this does not seem to fit reality nor align with the existing literature.
Additionally, the authors must take into account the following:
New Point 1: The abstract must be adapted to reflect the new analysis.
New Point 2: It is not clear how the intervals are constructed in each methodology. It is not necessary to model them, but it is essential to explain which values are taken as a reference.
New Point 3: In general, the structure of the paper is somewhat disorganized:
"Materials and Methods" is an appropriate name when a strictly empirical contribution is made. However, in this work, the contribution aims to be methodological, so an entire section should be dedicated to developing the proposed fuzzy time series approach. This section is not about materials but rather something like “Fuzzy Time Series and Proposed Method.”
The next section should focus on the empirical validation of the proposed method. It could present the materials (pension payments in Egypt) and methods (exemplification of the fit, comparison of point and interval predictions with other methods, and the metrics used).
Subsequently, the Results section should discuss the findings of the empirical application.

Experimental design

Point 7. It is necessary to discuss the approach used in comparison to those currently existing (whether fuzzy or not) and to highlight the potential theoretical and practical implications of the proposed fuzzy time series model.

This has been done. However, it is implemented in a disorganized manner. When developing the numerical application, it is necessary to first outline what analyses will be conducted (e.g., exemplification, comparison with other methods for point predictions and interval prediction, metrics used). Then, the database should be presented, followed by the development of the proposed analyses.

Validity of the findings

A significant issue in the paper is that the procedures commonly used in actuarial literature to predict pensions are not discussed. Additionally, the use of an additive modeling approach, instead of a multiplicative one—which is the standard in actuarial practice—is not justified.

---

## Round 0.3 · accepted · Accept

· Academic Editor

Accept

Dear authors,

Thank you for the revision. Your paper seems sufficiently improved and I think that necesseary additionss and modifications are correctly performed. Furthermore, in production stage, the variables in the figures should be written in italic as in the text. Please pay special attention to the usage of abbreviations. Spell out the full term at its first mention, indicate its abbreviation in parenthesis and use the abbreviation from then on.

Best wishes,

Reviewer 1 ·

Basic reporting

Authors have improved the paper throughout the revisions.

Experimental design
* * *
Validity of the findings
* * *
Additional comments
* * *
Reviewer 2 ·

Basic reporting

The revised version need proper English editing.

Experimental design

Improved well.

Validity of the findings

No comments.

Additional comments

The authors addressed all my comments.